# Optimal tagging strategies for illuminating expression profiles of genes with different abundance in zebrafish

Jiannan Liu[1], Wenyuan Li[1], Xuepu Jin[1], Fanjia Lin[1], Jiahuai Han [1,2,3✉] & Yingying Zhang [1✉]

CRISPR-mediated knock-in (KI) technology opens a new era of fluorescent-protein labeling in zebrafish, a preferred model organism for in vivo imaging. We described here an optimized zebrafish gene-tagging strategy, which enables easy and high-efficiency KI, ensures high odds of obtaining seamless KI germlines and is suitable for wide applications. Plasmid donors for 3′-labeling were optimized by shortening the microhomologous arms and by reducing the number and reversing the sequence of the consensus Cas9/sgRNA binding sites. To allow for scar-less KI across the genome, linearized dsDNA donors with 5′-chemical modifications were generated and successfully incorporated into our method. To refine the germline screen workflow and expedite the screen process, we combined fluorescence enrichment and caudal-fin junction-PCR. Furthermore, to trace proteins expressed at a low abundance, we developed a fluorescent signal amplifier using the transcriptional activation strategy. Together, our strategies enable efficient gene-tagging and sensitive expression detection for almost every gene in zebrafish.

[1] State Key Laboratory of Cellular Stress Biology, School of Life Sciences, Faculty of Medicine and Life Sciences, Xiamen University, 361102 Xiamen, Fujian, China. [2] Laboratory Animal Center, Xiamen University, 361102 Xiamen, Fujian, China. [3] Research Unit of Cellular Stress of CAMS, Cancer Research Center of Xiamen University, Xiang'an Hospital of Xiamen University, School of Medicine, Xiamen University, 361102 Xiamen, Fujian, China. ✉email: jhan@xmu.edu.cn; y.zhang@xmu.edu.cn

As a preferred model organism for in vivo imaging, zebrafish is used to resolve many scientific questions focused on developmental biology, regeneration, and neuronal circuits, etc.[1–6]. In vivo imaging requires fluorescence-labeling techniques, among which the CRISPR/Cas9-mediated knock-in (KI) technology is currently the most advanced technology to trace cells at in vivo levels[7–12]. CRISPR/Cas9-mediated KI achieves targeted insertion of exogenous DNA sequences in endogenous genes by DNA repair mechanisms such as homologous recombination (HR), non-homologous end joining (NHEJ), and microhomology-mediated end joining (MMEJ)[13,14]. However, HR-mediated KI shows low efficiency and difficulties in donor construction, restricting its application[15–18]; NHEJ-mediated intron targeting strategy has impressively high KI efficiency but it is not applicable to genes without introns[19–22]; MMEJ-mediated KI, in theory, should be very efficient but in practice variations in KI success rates using this method cannot be ignored[23–25]. Therefore, it is necessary to develop an optimized KI strategy through which fluorescent labeling of any gene can easily be achieved in zebrafish.

Difficulties exist in fluorescence detection when the tagged gene is endogenously expressed at a low abundance[26]. With the development of the CRISPR activation (CRISPRa) strategies, such as dCas9-VP64, SAM, SunTag, VPR, MPH, and SPH, fluorescent signals could be amplified by transcriptional activation of the tagged genes[26–34]. However, a direct increase of the endogenous expression of some genes, for instance, the *connexins*, is not only artificial but also detrimental, leading to abnormalities such as dysplasia, tumorigenesis, cancer cell migration, and cell death[35–38]. A recent report shows that the SPH-OminiCMV-Ents strategy enables fluorescent monitoring of low-abundance transcripts in vitro[26]. However, a method to trace low-abundance genes in vivo without interfering with the endogenous gene expression is still lacking.

Here, we reported an MMEJ-mediated KI method named S-NGG-25, when combined with a modified germline screen protocol, ensures easy manipulation, high KI efficiency, and high rates in obtaining desired seamless $F_1$ carriers, as evaluated by tagging 33 *connexin* genes in zebrafish. In addition, the S-NGG-25 method was successfully applied to genes that are hard to tag, such as *cx43.4*[25], and genes crucial for embryonic development, such as *tbx5a* and *tnni1b*[39–42]. To get rid of the unwanted plasmid backbone and achieve scar-less KI at any site across the genome, 5′-end chemically modified linearized dsDNA donors were employed. To track down low-expression proteins in zebrafish, we tagged low-expression genes with a fluorescence amplification system (the VH system), which amplifies fluorescent signals without interfering with the endogenous expression of the target genes. In sum, our method is optimal for KI and expression tracking of almost every gene in zebrafish.

## Results

**S-25, an optimized donor, ensures high-efficiency MMEJ-mediated KI**. The success rate of MMEJ-mediated KI varies from gene to gene and we failed to generate inheritable fluorescence-labeled *cx43* or *cx39.9* alleles using the reported GeneWeld strategy[25]. Junction PCR and sequencing analysis revealed reverse integration of the full-length linearized donor into the $F_1$ genome at the *cx43* locus (Supplementary Fig. 1), suggesting that the two consensus Cas9/sgRNA binding sites (*lamGolden*)[43] in the donor plasmid were not cut simultaneously. Instead, cutting at either site prohibited MMEJ- while facilitating NHEJ-mediated KI.

Besides using two cutting sites, a single-cut strategy was tested, but it failed to generate MMEJ-mediated KI zebrafish[23]. Since theoretically, a single cut should be more efficient, we generated a series of donors and tried to re-evaluate the KI efficiency of the single-cut method (Fig. 1a). The MMEJ-single donor contains only one artificial *lamGolden* Cas9/sgRNA site flanked by a left homologous arm and a right homologous arm. The MMEJ-double donor contains two *lamGolden* sites. A series of MMEJ-single donors and MMEJ-double donors were designed containing homologous arms of different lengths, 10 bp, 25 bp, 40 bp, or 100 bp (Supplementary Fig. 2a). To make a comprehensive comparison, HR and NHEJ donors were included[17,19,20] (Fig. 1a) and the target for GFP-tagging was *cx43* or *cx43.4*.

Donors were constructed in accordance with the high-efficiency sgRNA and were co-injected into one-cell-stage embryos with Cas9 mRNA, sgRNA for *cx43* or *cx43.4*, and *lamGolden* sgRNA. Same GFP patterns were detected in 'MMEJ-single' $F_0$ founders (Supplementary Fig. 2b, c), whereas random GFP expression was found in 'MMEJ-double' $F_0$ (Supplementary Fig. 2d). KI efficiencies were evaluated by the ratios of GFP-positive embryos at 48 h post fertilization (hpf). More KI embryos were obtained using the S-25 donor (an MMEJ-single donor with 25-bp homologous arms) than using the other donors (Fig. 1b, c). 5′- and 3′-junction PCR confirmed that KI occurred in S-25-donor-generated $F_0$ embryos and that KI happened more frequently in GFP-positive embryos than in GFP-negative ones (Supplementary Fig. 3a–c). Thus, MMEJ works better than HR and NHEJ in mediating KI in zebrafish and S-25 is the optimal donor for MMEJ-mediated KI.

To determine if the S-25-mediated GFP-tagging is inheritable, GFP-positive and GFP-negative $F_0$ were analyzed for germline transmission. For *cx43.4*, 3 out of 10 (30%) GFP-positive $F_0$ harbored transmittable KI alleles. $cx43.4^{+/+eGFP}$ $F_1$ showed GFP expression in the notochord, spinal cord, cerebral cortex, cornea, and hatching gland (Fig. 1d, e). For *cx43*, 3 out of 6 (50%) GFP-positive $F_0$ transmitted *cx43-eGFP* alleles to their $F_1$ progeny. $cx43^{+/+eGFP}$ $F_1$ showed GFP expression in the notochord, spinal cord, lens capsule, and ultimobranchial body (Fig. 1f, e). Junction PCR and sequencing for GFP-positive $F_1$ confirmed the germline transmission and also revealed that the mosaicism of the germline of the founders ranged from 12.3% to 31.2% for *cx43.4* and from 10.9% to 26.7% for *cx43* (Fig. 1g, Supplementary Fig. 4). As expected, no GFP-positive $F_1$ were observed from GFP-negative $F_0$ (Fig. 1e). Besides, microinjection-related injury and gene-editing-material-caused toxicity did not affect the establishment of the KI germline since the survival rates of the tested groups were all above 50% (Supplementary Fig. 5a, b).

**Evaluation of the S-25 strategy by tagging all zebrafish connexins**. To test whether our S-25 strategy can be applied to other genes, we tagged the rest 31 *connexin* genes with *GFP* or *mCherry*. However, fluorescence and germline transmission were only observed in $F_0$ and $F_1$ for *cx39.9* KI (Fig. 2a, Supplementary Fig. 6). *cx30.3*, *cx34.4*, *cx35*, *cx44.1*, *cx47.1*, *cx48.5*, *cx52.6*, and *cx55.5* were randomly selected to investigate reasons for the absence of fluorescence. KI actually occurred in microinjected $F_0$ of these *connexins* (Supplementary Fig. 7a), and these genes were expressed during development (Supplementary Fig. 7b, c), and sgRNA optimization could not solve the problem (Supplementary Fig. 7d). Thus, the lack of fluorescence in $F_0$ might be caused by the following reason(s): (1) the chimeric KI in $F_0$; (2) a dramatic decrease of the endogenous *connexin* expression by the labeling; (3) potential issues with GFP expression or detection.

Since chimeric KI in $F_0$ cannot truly reveal gene expression patterns, to obtain correctly whole-body-engineered KI zebrafish, we designed a workflow for efficient germline transmission screen integrating junction PCR and fluorescence observation (Supplementary Figs. 8–11 and Supplementary Data 1). Using this

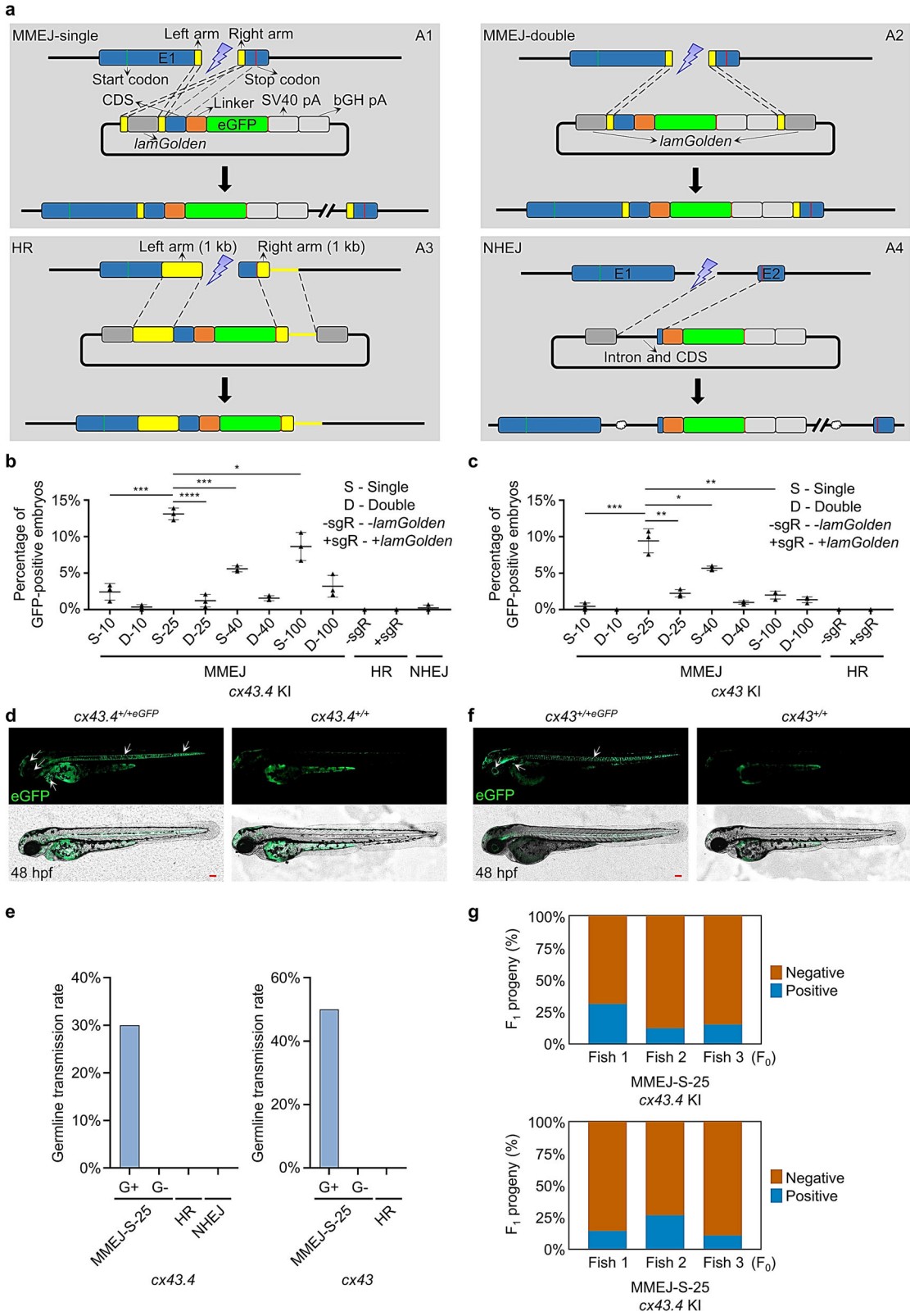

protocol, *eGFP*- or *mCherry*-tagged $F_1$ for all of the tested *connexins* were generated and at least two $F_0$ founders were obtained for each *connexin* (Fig. 2b). As expected, the cleavage efficiency of a given sgRNA positively correlates with the ratio of PCR-positive $F_0$ (Fig. 2c) and the chance of germline transmission positively links to the recombination verified by junction PCR of $F_0$ (Fig. 2d). The mosaicism of the germline of $F_0$ founders

was similar to that of *cx43.4* and *cx43*, as exemplified by *cx30.3* and *cx34.4* ranging from 5.8% to 35.4% (Fig. 2e). This method was then easily and successfully applied to all the rest 31 *connexins*. The lack of fluorescence in $F_0$ of *cx23*, *cx30.3*, *cx44.1*, and *cx48.5* was verified to be a result of the chimeric KI in the $F_0$ as revealed by the fluorescence-positive $F_1$ (Supplementary Fig. 10). Together, a combination of the S-25 donor and the

**Fig. 1 S-25 donor ensures efficient MMEJ-mediated KI in zebrafish. a** Schematic diagrams of MMEJ-, HR-, and NHEJ-mediated KI strategies, respectively. MMEJ-based donor vector containing a single artificial consensus Cas9/sgRNA binding site (*lamGolden*) was called MMEJ-single (A1), while the one with two *lamGolden* sites was named MMEJ-double (A2). One-kb homologous arms were used in the design for HR-based donors (A3). The NHEJ-based donor was shown in A4. Exons, *lamGolden* sites, linkers, *eGFP*, and SV40+bGH polyA are colored blue, gray, orange, green, and blank, respectively. Introns are shown as black lines. Homologous arms are shown in yellow. Start and stop codons are labeled as green and red vertical lines, respectively. **b**, **c** Percentages of GFP-positive embryos for *cx43.4* and *cx43* KI, respectively. Each donor was co-microinjected into wild-type (WT) zebrafish embryos at the one-cell stage with the Cas9/sgRNA system. GFP-positive embryos were counted at 48 h post fertilization (hpf). Donors are as follows: S-10, S-25, S-40, and S-100 donors are MMEJ-single with 10-bp, 25-bp, 40-bp, and 100-bp homologous arms, respectively; D-10, D-25, D-40, and D-100 donors are MMEJ-double with 10-bp, 25-bp, 40-bp, and 100-bp homologous arms, respectively; +sgR, microinjection with *lamGolden* sgRNA; -sgR, microinjection without *lamGolden* sgRNA. At least 100 embryos were analyzed for each group in one experiment. Data represent mean ± SD of 3 independent experiments. *P*-values were calculated using an unpaired Student's *t*-test. *$P < 0.05$; **$P < 0.01$; ***$P < 0.001$; ****$P < 0.0001$. **d**, **f** Images of $F_1$ zebrafish carrying KI alleles in *cx43.4* or *cx43* loci. White arrowheads indicate GFP signals in the notochord, spinal cord, cerebral cortex, cornea, and hatching gland of a *cx43.4*$^{+/+eGFP}$ larva at 48 hpf (**d**) and in the notochord, spinal cord, lens capsule, and ultimobranchial body of a *cx43*$^{+/+eGFP}$ larva at 48 hpf (**f**). Scale bars, 100 μm. Images are representatives of at least 10 larvae. **e** Germline transmission rates of GFP-tagged *cx43.4* or *cx43* using different strategies. "G+" represents GFP-positive $F_0$ zebrafish, and "G−" represents GFP-negative $F_0$ zebrafish. At least 6 $F_0$ were analyzed for each group. **g** Mosaicism of the germline of $F_0$ founders was determined by the percentage of $F_1$ carrying the KI cassette. GFP-positive *cx43.4*$^{+/+eGFP}$ or *cx43*$^{+/+eGFP}$ $F_1$ were in blue, and GFP-negative *cx43.4*$^{+/+}$ or *cx43*$^{+/+}$ $F_1$ were in orange. More than 100 $F_1$ were examined for each $F_0$ founder.

germline transmission screen workflow is an optimal protocol for high-efficiency KI in zebrafish.

**Reducing non-homologous residues introduced by *lamGolden* sequence increases the accuracy of MMEJ-mediated KI**. To achieve precise KI of a C-terminal tag, MMEJ-mediated precise repair at the 5′-junction is more critical than that at the 3′-junction. The *lamGolden* sequence is non-homologous with the target sequence decreasing the accuracy and efficiency of MMEJ-mediated repair and increasing the chances of random insertions and repair errors. Based on past experience, Cas9 usually cleaves at the site around 3 bp upstream of the protospacer adjacent motif (PAM). Thus, a sense (NGG) or an antisense (CCN) *lamGolden* site will generate a shorter (6 bp) or a longer (17 bp) 5′-overhang of the 5′-homologous arm after Cas9 cleavage. To test which overhang was better, we constructed donors containing a sense *lamGolden* (S-NGG-25) or an antisense *lamGolden* (S-CCN-25), respectively (Fig. 3a). S-NGG-25 and S-CCN-25 did not differ in producing GFP-positive $F_0$ for *cx43* or *cx43.4* (Fig. 3b), but S-NGG-25 gave higher chances of precise repair at 5′-junctions than S-CCN-25 as revealed by junction PCR and sequencing of $F_1$ (Fig. 3c, Supplementary Fig. 12a, b).

We also applied the S-NGG-25 method to *tbx5a* and *tnni1b*, two essential genes in development[39–42]. Survival rates of $F_0$ embryos were 30% for *tbx5a* and 15% for *tnni1b*. Abnormal $F_0$ was observed as reported[39,42] (Fig. 3d). Nevertheless, GFP-chimeric $F_0$ and desired KI $F_1$ exhibiting correct GFP patterns were obtained (at 48 hpf, *tbx5a* was expressed in the pectoral fin and heart while *tnni1b* was exclusively expressed in the heart) (Fig. 3e, f, Supplementary Fig. 13). Collectively, the S-NGG-25 method improves the efficiency and accuracy of MMEJ-mediated repair.

**Direct fluorescence tagging is suitable for visualizing proteins of high abundance**. It is worth noting that after the $F_1$ screen, fluorescent signals were only detectable for 7 out of 33 fluorescent-protein-tagged *connexins* (Supplementary Fig. 14). Since eGFP/mCherry was co-expressed with the endogenous gene, the absence of fluorescent signals in *cx34.4*$^{+/+eGFP}$, *cx35*$^{+/+eGFP}$, *cx47.1*$^{+/+eGFP}$, *cx52.6*$^{+/+eGFP}$, and *cx55.5*$^{+/+mCherry}$ $F_1$ (Supplementary Fig. 11) is very likely to have been resulted from low expression of the target gene. To test this speculation, correct inheritable KI was first confirmed by genotyping $F_2$ obtained from inbreeding fluorescence-negative $F_1$ carriers (Supplementary Fig. 15a, b). Then, the transcription of target genes in

fluorescence-negative wild-type, heterozygous, or homozygous $F_3$ was analyzed. qPCR results showed that fluorescent labeling did not dramatically decrease endogenous target gene expression (Supplementary Fig. 15c) and that the endogenous expression of *cx43* or *cx43.4* was much higher than that of other *connexins*, supporting that higher endogenous expression of target genes leads to stronger fluorescence signals (Fig. 4a). Thus, the most likely reason for the absence of fluorescence in KI zebrafish for the rest 26 *connexins* is probably the low expression level of the endogenous target genes.

**Incorporation of the Gal4-UAS signal amplification system to visualize gene expression in zebrafish**. To uncouple the fluorescence tag with the low-expressed endogenous genes, we designed an indirect fluorescence-labeling (IFL) strategy based on the Gal4-UAS and CRISPR activation system[28,44–47]. A transcriptional activator (TA) was fused to the target gene through MMEJ-mediated KI using an S-NGG-25 donor. Ideally, the TA co-expressed with the target gene will bind the originally silent promoter of eGFP (Supplementary Fig. 16), leading to amplified GFP expression. To optimize the original eGFP promoter (5×nrUAS-mini)[47], we constructed a series of eGFP reporters carrying mini promoters including *E1b*, *miniCMV*[26], *TATA* (random sequence containing *TATA* box)[48], and random sequence (without *TATA* box). Only the *miniCMV* promotor led to leaky expression of eGFP (Supplementary Fig. 17a, b), and the *5×nrUAS-E1b* promoter drove more efficient and stable eGFP expression than the others in the presence of the TA, Gal4-VP64 (Supplementary Fig. 17c, d). The TA was also optimized by constructing Gal4-VP64 (V), Gal4-VP64-P65 (VP), Gal4-VP64-HSF1 (VH), Gal4-VP64-P65-HSF1 (VPH), SPH (working with SunTag[29,34]), and SV (working with SunTag) plasmids and a combination of *5×nrUAS-E1b-eGFP* and Gal4-VP64-HSF1 gave far stronger fluorescence-signals than the other combinations (Supplementary Fig. 18).

Gal4-VP64, Gal4-VP64-HSF1, and Gal4-SPH TAs were then constructed respectively into S-NGG-25 donors carrying a *5×nrUAS-E1b-eGFP* so that all components of the signal amplification system could be integrated into the target gene as a whole (Fig. 4b). Direct fluorescence-labeling (DFL) donors were included as controls and *cx43.4* was chosen as the first target gene. GFP could be detected in all $F_0$, but the VH donor provided the strongest fluorescence in both $F_0$ and $F_1$ (Supplementary Fig. 19 and Fig. 4c) and generated GFP patterns comparable to that in DFL-embryos whereas embryos labeled by others showed certain variations (Figs. 1d and 4d, Supplementary Fig. 20).

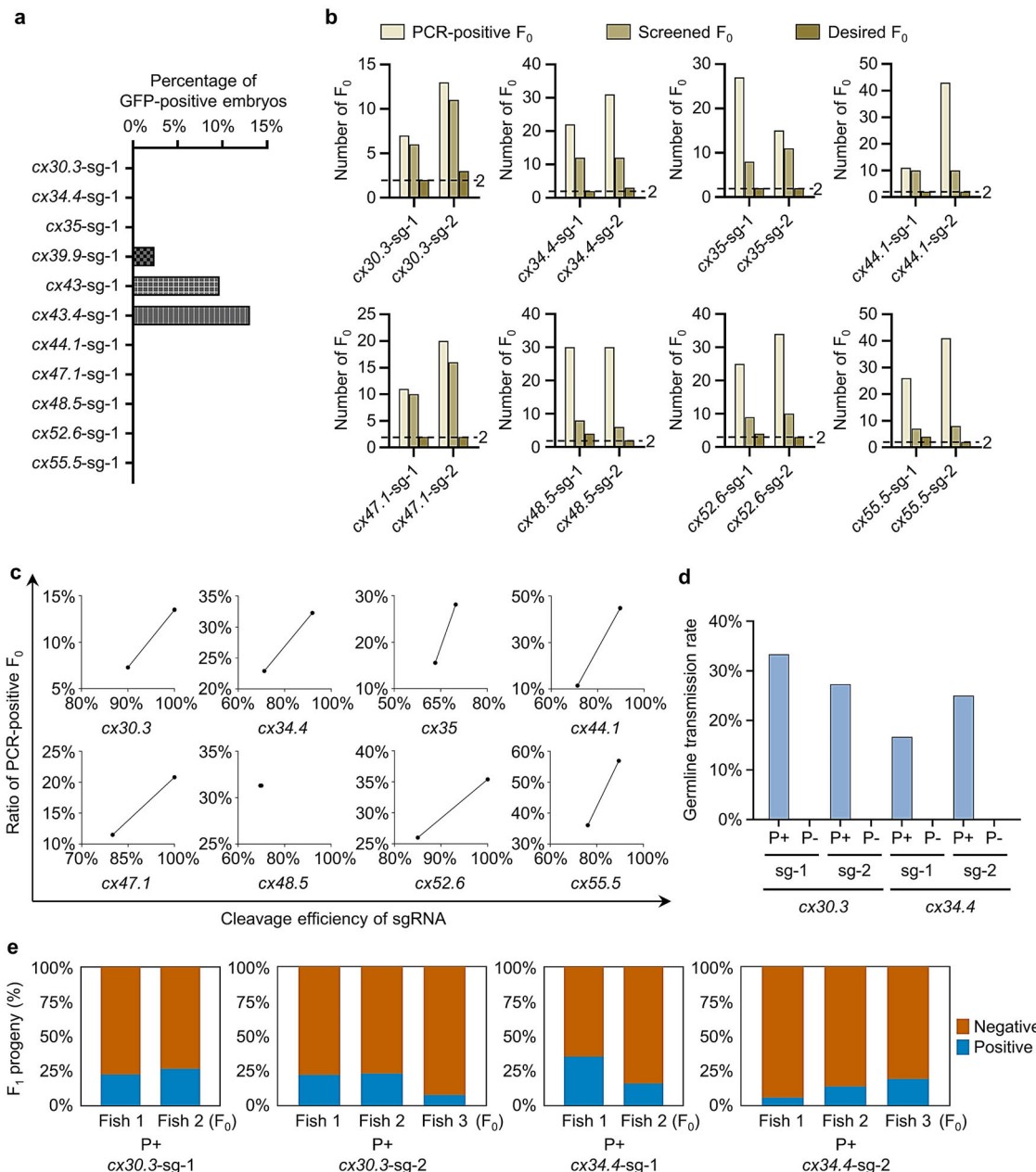

**Fig. 2 Evaluation of the S-25 strategy by tagging all zebrafish *connexins*. a** Percentages of GFP-positive $F_0$ embryos after tagging *cx30.3*, *cx34.4*, *cx35*, *cx39.9*, *cx43*, *cx43.4*, *cx44.1*, *cx47.1*, *cx48.5*, *cx52.6*, or *cx55.5*. S-25 strategy was performed and GFP-positive $F_0$ embryos were counted at 48 hpf. At least 200 embryos were analyzed for each gene. **b** Number of the desired $F_0$ for *cx30.3*, *cx34.4*, *cx35*, *cx44.1*, *cx47.1*, *cx48.5*, *cx52.6*, or *cx55.5* KI. Genomic DNAs were extracted from the caudal fins of 96 injected $F_0$ zebrafish (except for *cx55.5*, the genomic DNAs of which were from 72 $F_0$ zebrafish) and used for 5'-junction PCR. PCR-positive $F_0$ are those positive for 5'-junction PCR. Screened $F_0$ are the PCR-positive $F_0$ outbred with WT zebrafish. Desired $F_0$ are the screened $F_0$ whose $F_1$ offspring had precisely repaired 5'-junction. Two high-efficiency sgRNAs were chosen for each gene KI (sg-1 and sg-2). **c** Correlation between the cleavage efficiencies of sgRNAs and the ratios of PCR-positive $F_0$. The cleavage efficiency of a sgRNA was evaluated by PCR product enzymatic digestions at 24-h post injection or software analysis by the TIDE website. **d** Germline transmission rates of PCR-positive $F_0$ and PCR-negative $F_0$ for *cx30.3* and *cx34.4* KI. "P+", PCR-positive $F_0$; "P-", PCR-negative $F_0$. Two high-efficiency sgRNAs were used for KI at either locus. At least 6 $F_0$ were analyzed for each group. **e** Mosaicism of the germline of $F_0$ founders for *cx30.3* and *cx34.4* KI was determined by the percentage of $F_1$ carrying the KI cassette. PCR-positive *cx30.3*^{+/+eGFP} or *cx34.4*^{+/+eGFP} $F_1$ were shown in blue and PCR-negative *cx30.3*^{+/+} or *cx34.4*^{+/+} $F_1$ were in orange. More than 70 $F_1$ were examined for each PCR-positive $F_0$ founder.

Together, VH-mediated IFL provides dramatic signal amplification and correct expression patterns in zebrafish.

After proof-of-concept using *cx43.4*, we successfully applied the IFL method to low-expression genes, including *cx34.4*, *cx35*, *cx47.1*, *cx52.6*, and *cx55.5*, and easily detected strong and correct GFP signals in all IFL-labeled $F_1$ (Supplementary Fig. 21).

Expression patterns of *cx35* and *cx52.6* were consistent with the published in-situ hybridization data (ZFIN ID: ZDB-FIG-051103-56 and ZDB-FIG-060130-557), supporting the accuracy of the IFL method. Therefore, the VH-mediated IFL strategy is an optimal signal amplification method to tag most, if not all, low-expression genes in zebrafish.

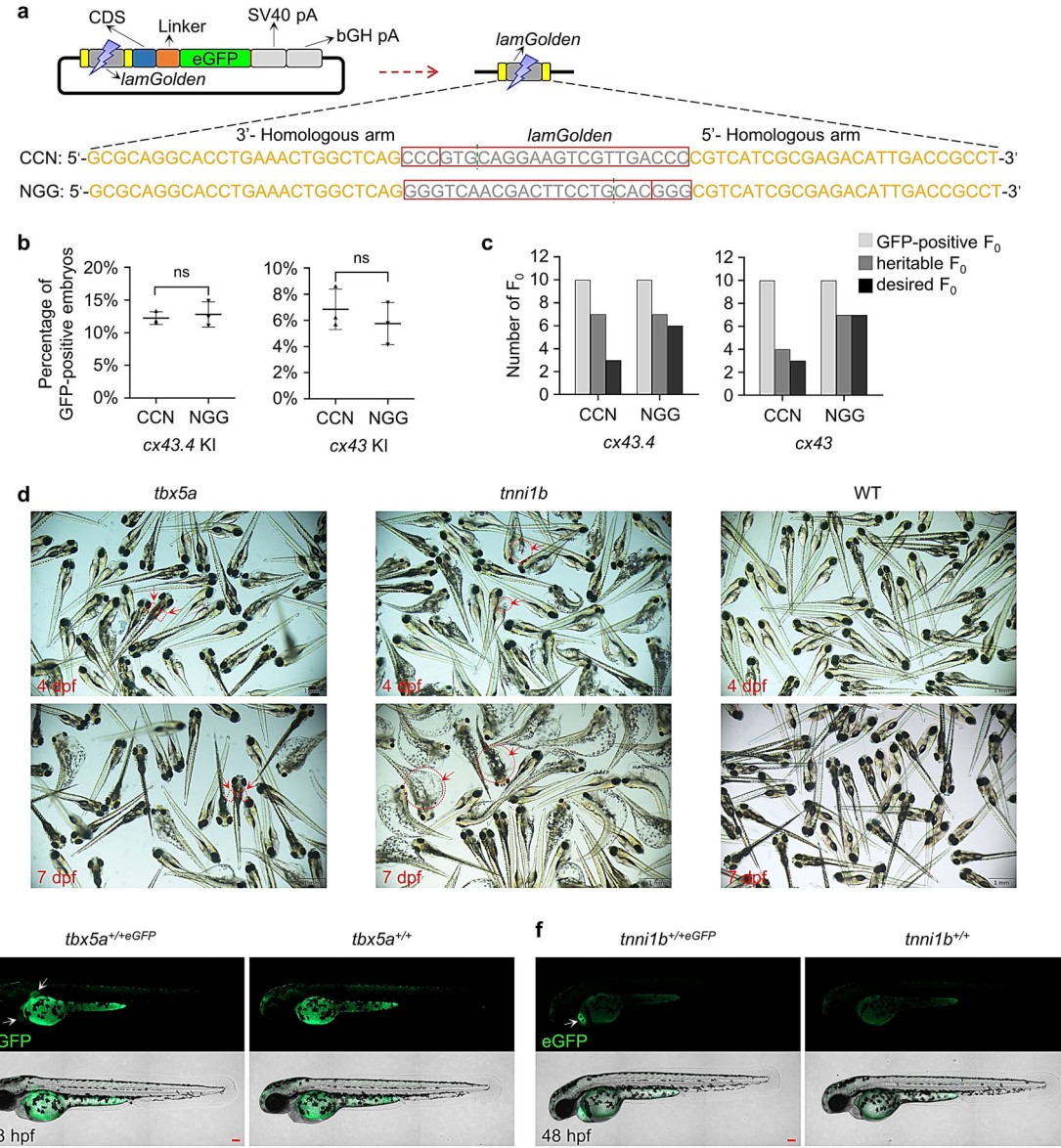

**Fig. 3 Reducing non-homologous residues introduced by *lamGolden* sequence increases the accuracy of MMEJ-mediated KI. a** A schematic diagram of S-NGG-25 and S-CCN-25 donors for *cx43.4* KI. Sense or antisense *lamGolden* site is represented by "NGG" or "CCN". Predicted cleavage sites of Cas9 were indicated by green dotted lines, and residuals left in homologous arms were shown in gray. **b** Percentages of GFP-positive embryos obtained by using S-CCN-25 or S-NGG-25 strategy for *cx43.4* or *cx43* KI. GFP-positive embryos were counted at 48 hpf. At least 100 embryos were analyzed for each strategy for each gene. Data represent mean ± SD of 3 independent experiments. *P*-values were calculated using an unpaired Student's *t*-test. ns, not significant. **c** Numbers of desired $F_0$ obtained using S-CCN-25 or S-NGG-25 strategy. Heritable $F_0$ are the GFP-positive $F_0$ whose KI alleles can be detected in their $F_1$. Desired $F_0$ are the GFP-positive $F_0$ whose KI alleles were verified in their $F_1$ to be heritable and precise by 5′-junction PCR. **d** Images of $F_0$ embryos after *tbx5a* or *tnni1b* KI. Cas9 mRNA, sgRNA (targeting *tbx5a* or *tnni1b*), and the corresponding S-NGG-25 donor were co-injected into WT one-cell-stage embryos; images were taken at 4 and 7 days post fertilization (dpf). Deficient embryos were indicated by red arrowheads. WT embryos were used as the control. Scale bars, 1 mm. **e**, **f** Images of *tbx5a*$^{+/+eGFP}$ and *tnni1b*$^{+/+eGFP}$ $F_1$. White arrowheads indicate GFP signals in the heart and pectoral fin of a *tbx5a*$^{+/+eGFP}$ larva at 48 hpf (**e**) and in the heart of a *tnni1b*$^{+/+eGFP}$ larva at 48 hpf (**f**). Scale bars, 100 μm. Images are representatives of at least 10 larvae.

**5′-end chemically modified linearized dsDNA for scar-less MMEJ-mediated KI.** To simplify the donor preparation process and also avoid plasmid backbone integration[49–51], linearized dsDNA donors containing functional cassettes were designed (Supplementary Fig. 22) and applied to *cx43* and *cx43.4*. However, no GFP-positive embryos were observed, leading to our speculation that the linearized donors were quickly degraded before MMEJ. dsDNA 5′-end chemical modifications were reported to resist degradation by exonucleases in cells[52–55]. Consistently, using a *CMV-eGFP* mini reporter, we proved that 5′-four phosphorothioate (5′-4PS)-modified[56] dsDNA gave

significantly higher GFP expression than the unmodified linearized dsDNA, while the circular dsDNA provided the highest GFP expression, supporting that the circular dsDNA was more stable in cells than the linearized dsDNAs (Supplementary Fig. 23a–c).

Optimizations were done by evaluating more 5′-chemical modifications, including 8PS, 12PS, 24PS, C6, C12, biotin[57], spacer18, and acrydite[57] (Supplementary Fig. 23d), in *cx43.4* and *cx43* KI based on the S-NGG-25 method (Fig. 5a). All of the 5′-modified dsDNA donors led to more GFP-positive $F_0$ embryos than the unmodified donor, and the 5′-12PS and 5′-C6 dsDNAs generated the most GFP-positive $F_0$ embryos (Fig. 5b, c). Circular

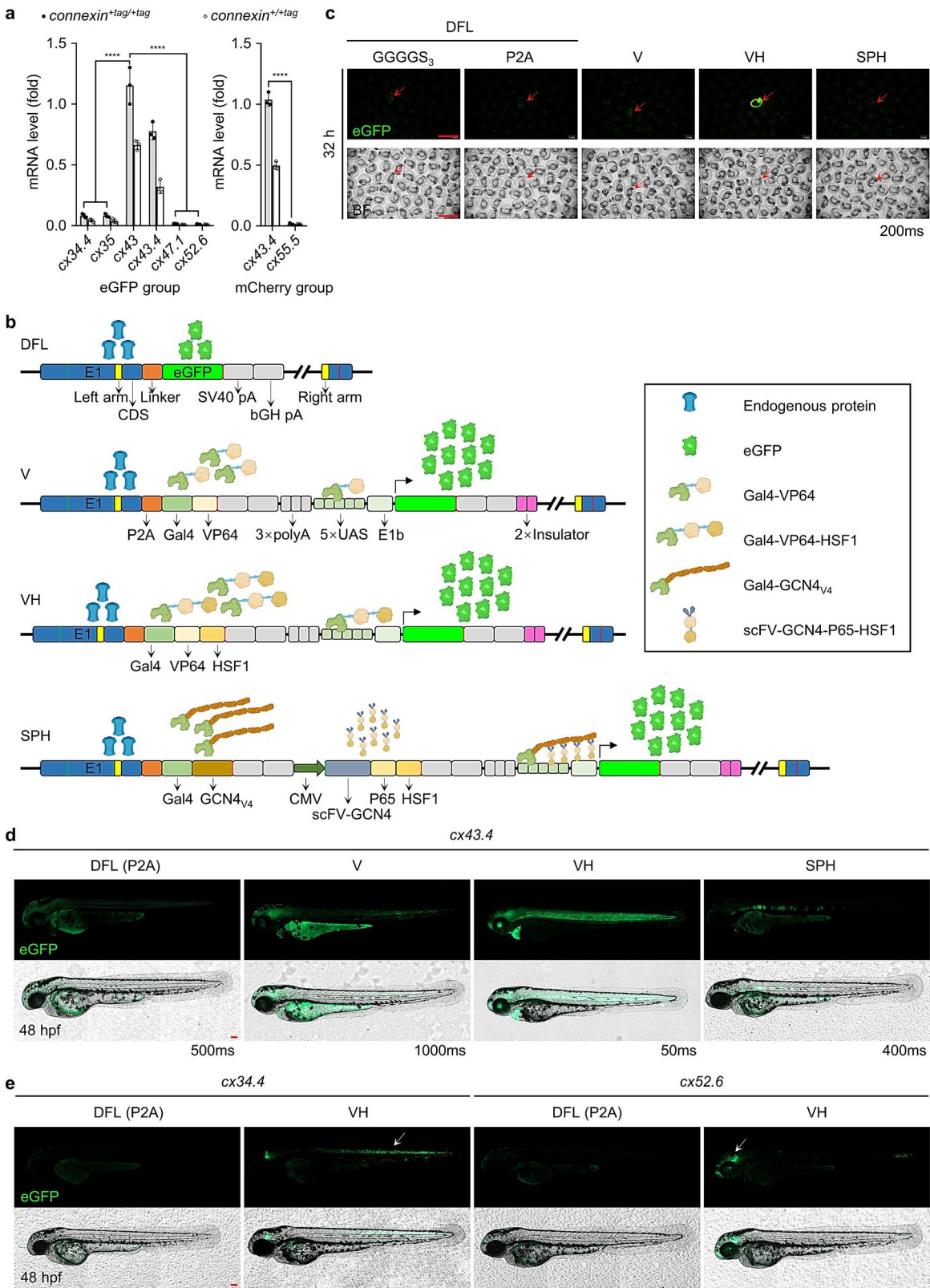

donors worked better than linearized donors, probably due to the fact that circular dsDNA has a longer half-life in cells (Supplementary Fig. 24). Precise repair and scar-less KI in GFP-positive $F_1$ generated by using the 5′-12PS donors was verified (Supplementary Fig. 25a, b). Using the 5′-12PS-dsDNAs, the germline transmission efficiency for GFP-labeled $cx43.4$ and $cx43$ alleles was about 80% and 75%, respectively, higher than that

using circular dsDNAs (Figs. 1e and 5d), and no germline transmission was observed in all GFP-negative $F_0$ (Fig. 5d). The mosaicism of the germlines ranged from 14% to 36% (Supplementary Fig. 25c). GFP patterns in 5′-12PS-mediated-KI $F_1$ for both $cx43.4$ and $cx43$ were the same as that generated by using circular donors (Figs. 5e, f and 1d, f). Precise and inheritable KI was further confirmed by genotyping $F_2$ generated by $F_1$

**Fig. 4 Fluorescent signals can be dramatically amplified by the VH strategy. a** mRNA levels of fluorescence-labeled *connexins* in $F_3$ zebrafish. Thirty *connexin*$^{+tag/+tag}$ or *connexin*$^{+/+tag}$ zebrafish larvae were pooled to extract RNA for qPCR. qPCR primers were designed to target the fluorescence tag. For the group with eGFP-labeling, data were normalized to *actb1* and relative to *cx43*$^{+eGFP/+eGFP}$. For the mCherry group, data were normalized to *actb1* and relative to *cx43.4*$^{+mCherry/+mCherry}$. Data are presented as mean ± SD of 3 independent experiments. Three repeats were included per group. *P*-values were calculated using an unpaired Student's *t*-test. ****$P < 0.0001$. **b** Schematic diagrams of the genome after editing by direct fluorescence-labeling (DFL), V, VH, and SPH strategies, respectively. All strategies are built on the S-NGG-25 KI method. Theoretically, for the DFL strategy, eGFP is directly linked to the target gene by a P2A linker or a GSSSS-linker and would be co-expressed with the target gene. For the V or VH strategy, Gal4-VP64 or Gal4-VP64-HSF1 would be co-expressed with the target gene, be separated from the target protein due to the function of a P2A peptide, and then induce eGFP expression by binding 5×nrUAS. For the SPH strategy, Gal4-GCN4 would be co-expressed with the target gene, be separated from the target protein, and then work together with scFV-GCN4-P65-HSF1, which is constitutively expressed in all cells, to transcriptionally activate expression of eGFP. **c** Fluorescence images of $F_1$ embryos carrying fluorescence-labeled alleles generated by using DFL, V, VH, or SPH strategy at the *cx43.4* loci at 32 hpf. Desired $F_1$ were screened out based on GFP expression patterns and genotyping results. Images were representatives of at least 10 desired $F_1$ and were captured under the same exposure time (200 ms). Scale bars, 2 mm. **d** GFP expression patterns in $F_1$ larvae generated by DFL, V, VH, or SPH strategy at the *cx43.4* locus. GFP was detected in the entire notochord of $F_1$ generated by DFL or VH but was only partially in the notochord of $F_1$ generated by V or SPH at 48 hpf. Exposure time was determined by the fluorescence intensity of GFP signals. Scale bars, 100 μm. Images are representatives of at least 10 $F_1$ larvae. **e** Images of $F_1$ larvae for *cx34.4* or *cx52.6* KI generated by a DFL strategy using a P2A linker or a VH strategy. GFP signals were detected in the spinal cord and hindbrain at 48 hpf in *cx34.4*$^{+/+VH}$ embryos and in the forebrain, midbrain, and posterior notochord at 48 hpf in *cx52.6*$^{+/+VH}$ embryos. Scale bars, 100 μm. Images are representatives of at least 10 $F_1$ larvae.

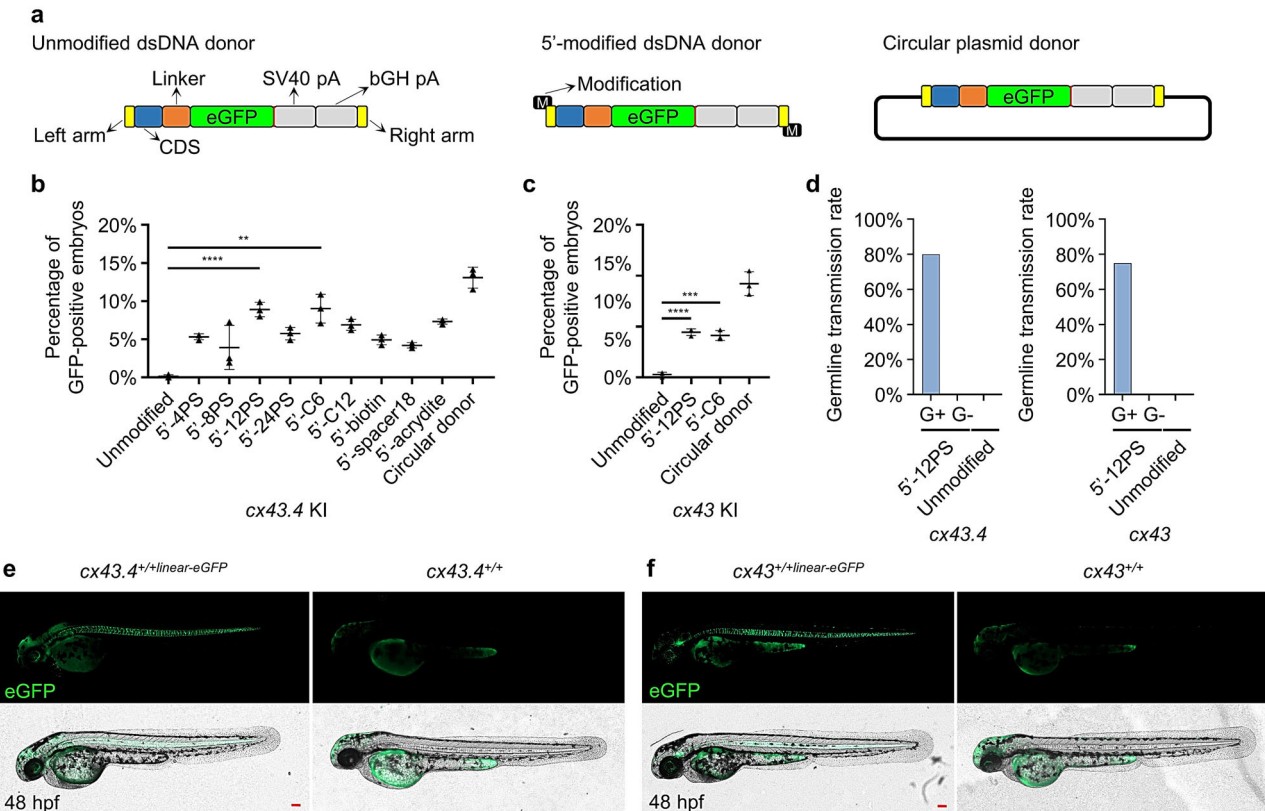

**Fig. 5 Establishment of 5′-end-modified dsDNA mediated KI system based on S-NGG-25 KI strategy. a** A schematic diagram of unmodified linearized dsDNA, 5′-modified linearized dsDNA, and circular dsDNA donors. All of the donors have the same integration cassette, including two homologous arms, a CDS, a linker, an eGFP coding sequence, and two polyA signals. 5′-modifications were added to the 5′-terminus of the linearized dsDNAs by using modified PCR primers. **b, c** Percentages of GFP-positive embryos after 1-cell-stage microinjection of Cas9/sgRNA together with different 5′-modified linearized dsDNA, unmodified linearized dsDNA, or circular dsDNA donors. *cx43.4* and *cx43* loci were chosen to evaluate modifications including PS, C6, C12, biotin, spacer18, and acrydite. GFP-positive embryos were counted at 48 hpf. At least 100 embryos were analyzed for each condition. Data represent mean ± SD of 3 independent experiments. *P*-values were calculated using an unpaired Student's *t*-test. **$P < 0.01$; ***$P < 0.001$; ****$P < 0.0001$. **d** Germline transmission rates of tagged *cx43.4* or *cx43* using the 12PS-modified linearized donors. "G+" represents GFP-positive $F_0$, and "G−" represents GFP-negative $F_0$. At least 4 GFP-positive or 20 GFP-negative $F_0$ embryos were raised to adulthood and outbred with WT zebrafish. The number of $F_0$ that can generate the desired $F_1$ was counted to calculate the germline transmission rate. **e, f** Images of *cx43.4*$^{+/+linear-eGFP}$ (**e**) and *cx43*$^{+/+linear-eGFP}$ (**f**) $F_1$ embryos derived from outbreeding the desired $F_0$. Images were taken at 48 hpf. Scale bars, 100 μm. Images are representatives of at least 10 GFP-positive $F_1$ embryos.

inbreeding (Supplementary Fig. 25d). To check the possibilities of random integration of the chemically modified linearized dsDNA, we applied an updated flanking sequence walking method, the Cyclic Digestion and Ligation-Mediated PCR (CDL-PCR) method[58], to the $cx43^{+linear-eGFP/+linear-eGFP}$ and $cx43.4^{+linear-eGFP/+linear-eGFP}$ KI zebrafish. The results showed that the modified linearized dsDNAs were integrated into the desired target sites for both genes, and no random integration was detected (Supplementary Fig. 26). Thus, the 5′-chemically modified linearized donors are excellent for efficient and accurate MMEJ-mediated KI.

## Discussion
Various KI strategies have been developed and applied in zebrafish to visualize endogenous gene expression and dynamics[17,19,23]. To supplement the current gene-editing toolbox, here, through several optimizations, we reported an improved MMEJ-based KI strategy for efficient and accurate fluorescence-labeling of almost every gene of interest in zebrafish, whether it is endogenously highly expressed or not.

In comparison with current MMEJ-mediated KI strategies, including the GeneWeld[11,23–25], the method reported here has several improvements. First, by reducing the number of Cas9/sgRNA binding sites from two to one, shortening the homologous arms to 25 bp, and reversing the sequence of the Cas9/sgRNA binding site from CCN to NGG, we generated an S-NGG-25 donor, which indeed provided higher efficiency and accuracy of the improved MMEJ-mediated KI; germline transmission efficiency and accuracy were further increased by the usage of 5′-modified linearized dsDNA donors. Second, the KI procedures introduced here were, in practice, easier and faster: (1) the donor construction process is much easier and faster using the S-NGG-25 method than the current methods since components, except for the CDS, can be incorporated into the donor backbone beforehand, allowing for convenient construction of multiple donors with the same tag simultaneously; (2) the 5′-modified linearized dsDNA donors are easily prepared without the need to clone in vivo linearization sites and can even be cloning-free; (3) the germline transmission screen workflow described here sets an example of accumulating inheritable KI alleles at the lowest cost of time and efforts. Third, the method reported here was verified to achieve broader applications: (1) in comparison to the NHEJ-mediated method, which is restricted to genes containing introns[19,20] and the GeneWeld, which could interfere with the expression of the target gene[25], the S-NGG-25 method is theoretically suitable for C-terminal tagging for all genes because the gRNA target site could be anywhere before the stop codon. Indeed, it was successfully applied to 33 *connexins*, including those hard to tag using current methods[25] (Fig. 1d). It was also verified to be applicable to genes essential for development, such as *tbx5a* and *tnni1b* (Fig. 3e, f); (2) using 5′-chemically modified linearized dsDNA donors as a complement to the S-NGG-25 method, integration of the plasmid backbone was avoided, and efficient, precise, and scar-less KI was achieved, expanding the application from C-terminal tagging to tagging in any region of the genome; (3) in combination with the Gal4-UAS signal amplification system, the S-NGG-25 method was validated to be applicable to various genes endogenously expressed at different abundances.

Using 5′-AmC6-modified linearized dsDNA as donors was recently reported by Andersson et al., which yielded high KI efficiency with short homologous arms[59]. Among the 5′-modifications we tested, 5′-AmC6 (5′-C6) was one of the better ones, consistent with the data by Andersson et al.

The CRISPR activation system was reported to significantly amplify the transcription of low-abundance genes in cell cultures[26] but it did not work well when applied to zebrafish. The reason is unknown but we speculated that the large size of the plasmid inhibited the process. On the contrary, the Gal4/UAS system, a widely used tool to regulate gene expression in zebrafish, was successfully integrated into our method to amplify fluorescent signals and visualize low-abundance genes.

To further explore our method, we tried to combine the VH strategy and the 5′-modified-linearized-donor-mediated KI strategy to generate KI zebrafish for *cx34.4*, *cx43.4*, and *cx47.1* (Supplementary Fig. 27). However, it turned out that the ratios of PCR-positive $F_0$ were very low, even though several PCR-positive $F_0$ embryos were obtained for each gene (Supplementary Fig. 27b). For *cx34.4* and *cx43.4*, junction PCR and GFP patterns confirmed that the 12PS-modified linearized dsDNA was precisely integrated into the target site and the VH-eGFP-tagged $F_1$ were generated (Supplementary Fig. 27c). But for *cx47.1*, no VH-eGFP-tagged $F_1$ zebrafish were obtained from 10 PCR-positive $F_0$. Thus, the 5′-chemically modified-linearized VH-adapted-donor-mediated KI strategy was feasible but the KI efficiency was low, and further optimizations were needed if it was to be applied to all genes.

Besides, other optimizations had been tested, such as increasing HDR (homology-directed repair) and inhibiting NHEJ repair. CtIP, the CtBP (C-terminal Binding Protein) interacting protein, is a protein involved in the early steps of HDR[60–62]. Researchers showed that fusion of the CtIP N-terminal 296 aa fragment (designated as HE) to Cas9 increased HR-dependent KI efficiency in human cell lines, iPS cells, and rat zygotes[63]. However, no improvement in KI efficiency in zebrafish was observed in our lab when either murine or human HE was fused to nzCas9 (a zebrafish codon-optimized Cas9) (Supplementary Fig. 28a). SCR7, an inhibitor of NHEJ repair, and RS-1, a stimulator of HDR[64–67], were also tested by us but neither of them improved KI efficiency or eliminated NHEJ repair in zebrafish (Supplementary Fig. 28b). Thus, these strategies may not be applicable to zebrafish.

In short, we systematically optimized and developed a KI strategy optimal for tagging almost every gene with different abundance in zebrafish. It provides a valuable tool for studying development, physiology, and other biological questions in zebrafish. It needs to be noted that a gene with an exogenous tag might influence its protein expression level and subcellular localization, which may need to be determined in the course of studies. An additional note for using our method is that caution is needed if the target gene expression is tightly regulated by its 3′ UTR because we incorporated an artificial 3′ UTR and polyA sequence in the donor vector.

## Methods
**Zebrafish husbandry**. Zebrafish, including adults and larvae, were maintained through standard protocols at 28.5 °C in the zebrafish aquarium system of Xiamen University with a 13/11 light/dark cycle. Zebrafish embryos were kept in Embryo Buffer (60 mM NaCl, 0.67 mM KCl, 1.9 mM NaHCO₃, 0.9 mM CaCl₂, 0.0002% Methylene blue) at a temperature (28 ± 0.5 °C) and light (13 h light/11 h dark cycle) controlled incubator. Feeding and general monitoring of all zebrafish were performed twice a day (9 A.M. and 4 P.M.). The wild-type strain used in this research was Tübingen (Tü). All zebrafish husbandry and experiments were reviewed and approved by the Laboratory Animal Management and Ethics Committee of Xiamen University and were in strict accordance with good animal practice as defined by Xiamen University Laboratory Animal Center. We have complied with all relevant ethical regulations for animal use.

**Plasmid construction and purification**. Plasmids were constructed by ligation-independent cloning (LIC)[68]. Plasmid

sequences are provided in Supplementary Data 2. For MMEJ-mediated KI donors, microhomologous arms were designed according to the sequence flanking the sgRNA target site in the genome. The complementary CDS was cloned from the genome and was the sequence from the cleavage site, which must be synonymously mutated to prevent cleavage by Cas9/sgRNA to the stop codon. Before microinjection, plasmids were purified using MinElute PCR Purification Kits (QIAGEN, 28004) following the manufacturer's instructions. All operations should be protected from RNase.

**Preparation of sgRNAs and zCas9 mRNA**. sgRNAs were designed using CRISPRscan (https://www.crisprscan.org/)[69]. sgRNA sequences are provided in Supplementary Data 3. A universal reverse primer (5′-AAAAAAAGCACCGACTCGGTGCCAC-3′) and a forward primer, which consists of successive sequences of a T7 promotor, a sgRNA target site, and a part of a sgRNA scaffold, were designed for sgRNA template synthesis through PCR from the pUC19-scaffold (5′-TAATACGACTCACTATAGG-GAATTGTC-CATTCCCCCA-GTTTTAGAGCTAGAAATAGC-3′, the first nucleotide of sgRNA needs to be replaced by a guanosine) (Chang et al.)[70]. PCR products were purified by MinElute PCR Purification Kits and then used for in vitro sgRNA transcription using T7 RNA polymerases (Vazyme, TR101-01). sgRNAs were purified by LiCl precipitation. The zebrafish codon-optimized Cas9 expression vector pTST3-nzCas9 was linearized by XbaI and used as the template for zCas9 mRNA in vitro transcription using the mMESSAGE mMA-CHINE™ T3 Transcription Kit (Invitrogen, AM1348). zCas9 mRNA was then purified by LiCl precipitation. All operations should be protected from RNase.

**Microinjections of zebrafish embryos**. To test sgRNA efficiency, zCas9 mRNA and sgRNA were co-injected into one-cell-stage zebrafish embryos. Each embryo was injected with 1 nL of the injection solution containing 300 ng/μL zCas9 mRNA and 50 ng/μL sgRNA. To perform KI, zCas9 mRNA, sgRNAs, and the corresponding donor plasmid were co-injected into one-cell-stage zebrafish embryos. Each embryo was injected with 1 nL of the injection solution containing 300 ng/μL zCas9 mRNA, 50 ng/μL for each sgRNA, and 20 ng/mL donor plasmid. Injection solutions should be injected into the animal pole of an embryo. Micro-injected zebrafish embryos were kept in Embryo Buffer at 28.5 °C in an incubator.

**Evaluation of sgRNA efficiency**. To extract genomic DNA, fifteen zebrafish embryos injected with zCas9 mRNA and sgRNA were divided into three groups, lysed in 50 mM NaOH solution (50 μL for each group) at 95 °C for 15 min, and then neutralized with 1 M Tris-HCl (pH = 8.0, 10 μL for each group). WT zebrafish genomic DNAs were used as negative controls. One μL of the genomic DNA was used for PCRs using primers flanking the target site (Supplementary Data 3). PCR products were sequenced or examined by enzymatic reactions. Sequencing data files (.ab1 format) were analyzed by the TIDE Web tool (https://tide.nki.nl/)[71] for indel frequencies.

**Calculation of GFP-positive or PCR-positive embryos after microinjection**. At 24 hpf (or 48 hpf as indicated), dead embryos were removed and GFP-positive embryos were counted under a stereomicroscope (Olympus, SZX16). Percentages of GFP-positive embryos in total living embryos were calculated. For PCR analysis, 1-month-old microinjected zebrafish were anesthetized with 0.02% tricaine (ethyl 3-aminobenzoate methane-sulfonate salt in 1 M Tris-HCl, pH 8.0). For each zebrafish, a small piece of the caudal fin was collected into a labeled PCR tube, and the corresponding anesthetized zebrafish were kept in UV-sterilized filtered water in one well of a six-well plate with the same label. The alkaline lysis method was used for genomic DNA extraction, followed by 5′-junction PCR. Percentages of PCR-positive larvae in the total analyzed larvae were calculated.

**Junction PCR, insertion PCR, and sequencing**. The alkaline lysis method was performed to extract genomic DNA used for short fragment PCR (<1 kb). For long fragment PCR (>1 kb), lysis buffer (10 mM Tris-HCl, pH 8.2, 200 mM NaCl, 5% SDS solution, 200 mg/mL proteinase K, and 10 mM EDTA) was used for genomic DNA extraction followed by ethanol precipitation. Lysis buffer extraction was also applied to embryo pools to get high-quality genomic DNA. Junction PCRs were then performed using 2×Taq Master Mix (Dye Plus) (Vazyme, P112 or 213) and primers flanking the microhomologous arms. Primers flanking the engineered region and 2×Phanta Max Master Mix (Dye Plus) (Vazyme, P525) were used for insertion PCR. Primers were provided in Supplementary Data 3. PCR products were subject to sequencing analysis.

**Defining screened, heritable, and desired $F_0$**. GFP-positive or PCR-positive embryos that were raised to adulthood and outbred with WT zebrafish were defined as screened $F_0$. $F_1$ embryos obtained from these screened $F_0$ were examined for GFP expression patterns and genotyped. If target insertion occurred in $F_1$ (whether perfect or not), the corresponding $F_0$ was defined as heritable $F_0$. If the insertion was accurate, the corresponding $F_0$ was defined as desired $F_0$.

**Imaging and processing**. At 24 hpf or 48 hpf, zebrafish embryos were anesthetized using 0.02% tricaine (ethyl 3-aminobenzoate methanesulfonate salt in 1 M Tris-HCl, pH 8.0), fixed in 0.8% low melting agarose gel (Sangon, A600015) in 35 mm confocal dishes (Avantor, VWR75856-740), and then imaged by fluorescence microscopy (Zeiss, Axio Observer 7). The objective was set to 10×, tile scan was set to 3 × 3, and Z-stack was set to 3.5 μm. Images were processed using ZEN 2.5 pro imaging software. Extended depth of field algorithm was used to generate a best focus composite image from a Z-stack.

**Real-time qPCR analysis**. Thirty zebrafish embryos of the same genotype were collected at indicated time points and used for total RNA extraction using RNAIso Plus reagent (TAKARA, 9109). The concentration of each RNA sample should be above 300 ng/μL. HiScript II Q RT SuperMix (+gDNA wiper) (Vazyme, R223) was used for reverse transcription. ChamQ Universal SYBR qPCR Master Mix (Vazyme, Q711) was used for qPCR following the manufacturer's instructions. Technical triplicates were performed for each cDNA template. Primers were designed near the 3′UTR and were provided in Supplementary Data 3. RT-qPCRs were performed by a Roche LightCycler 96. β-actin was used as the internal reference gene, and the target and reference genes had similar amplification efficiencies. Expression of the target gene was normalized to that of β-actin to compensate for any difference in the concentration of samples in every run of qPCR. The threshold cycle ($2^{-\Delta\Delta Ct}$) method was used and the fold change of the target gene in each sample relative to the biological control sample was plotted.

**Calculation for mean fluorescence intensity (MFI)**. After microinjection in one-cell-stage zebrafish embryos, dead embryos were removed at 8 hpf, and live embryos were gathered under a stereomicroscope (Olympus, SZX16). Cellsens software was used to capture images. The same exposure time was applied to all

images (200 ms). For each zebrafish strain, at least 60 embryos were imaged and analyzed. Cellsens was used to process images and to calculate the mean fluorescence intensity.

**Linearized dsDNA preparation and modification.** Modifications (PS, $NH_2C6$, $NH_2C12$, biotin, spacer18, or acrydite) (Supplementary Data 6) were added to the 5′ terminal of PCR primers (Sangon) which were then used to amplify the functional cassettes from the S-NGG-25 plasmid. PCR products were verified by gel electrophoresis and sequencing analysis. MinElute PCR Purification Kit was then used to purify the PCR products. All operations should be protected from RNase.

**Statistics and reproducibility.** The number of replicates for each experiment is indicated in the respective legends. In brief, to calculate the percentages of GFP-positive embryos, calculate survival rates of microinjected embryos, analyze the values of RT-qPCR results, and calculate the mean fluorescence intensity, three independent experiments were performed to ensure robustness and reproducibility. Statistical analyses were performed using GraphPad Prism (8.0.1). All plots represent the mean ± standard deviation (SD) of different biological samples. $P$-values were calculated using an unpaired Student's $t$-test. Differences were declared statistically significant if $P < 0.05$ and the following statistical significance indicators were used: $*P < 0.05$; $**P < 0.01$; $***P < 0.001$; $****P < 0.0001$.

**Reporting summary.** Further information on research design is available in the Nature Portfolio Reporting Summary linked to this article.

## Data availability

The authors declare that all data supporting the findings of this manuscript are available within the paper figures and supplementary information files (including supplementary figures and supplementary data). The source data for all graphs are provided as compiled Excel files (filename: Supplementary Data 4 and 5). Uncropped gels are shown in Supplementary Fig. 29. All other data are available from the corresponding author on reasonable request.

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

## Acknowledgements

The authors thank Bo Zhang (School of Life Sciences, Peking University, China) for providing Tübingen (Tü) wild-type zebrafish; Hui Yang (Shanghai Institutes for Biological Sciences, Chinese Academy of Sciences, China) for providing CRISPR activation-associated plasmids; and Dingyang Yuan (College of Bioscience and Biotechnology, Hunan Agricultural University, China) for providing the pMD18T-APS plasmid for the CDL-PCR. This work was supported by the National Key R&D Program of China (2020YFA0803500 to J.H.), the National Natural Science Foundation of China (82388201 to J.H.; 31801158 to Y.Z.), the CAMS Innovation Fund for Medical Sciences ((2019-I2M-5-062) to J.H.), the Fujian Province Central to Local Science and Technology Development Special Program (2022L3079 to J.H.), and the Fu-Xia-Quan Zi-Chuang District Cooperation Program (3502ZCQXT2022003 to J.H.). The funders had no role in study design, data collection and analysis, decision to publish, or preparation of the manuscript. We thank Lu Zhou (Xiamen University) for proofreading and editing the manuscript.

## Author contributions

J.L., W.L., and Y.Z. designed the experiments; J.L. and X.J. performed most of the experiments; F.L. participated in genotyping and zebrafish husbandry; J.L., W.L., and Y.Z. analyzed data; J.L., W.L., Y.Z., and J.H. wrote the manuscript; J.H. and Y.Z. conceived the project, designed the experiments, and supervised the study.

## Competing interests

The authors declare no competing interests.
