## [Peer Review File · Communications Biology]

Reviewers' comments:

Reviewer #1 (Remarks to the Author):

In the manuscript, the authors performed experiments to knock-in donor DNAs to the zebrafish genome via the CRISPR/Cas9 method. They tested many conditions to optimize the frequency. Namely, the number of cutting sites, the length of the homology arm, signal amplification via Gal4-UAS system, modification of the linearized donor, etc. Also, the authors applied this system to create KI in authors Cx genes and demonstrated this approach is feasible. Although the work itself is not perfectly innovative, the authors conducted comprehensive analyses and the information provided is useful to all researchers who are planning to do similar KI experiments.

I have the following comments:

(1) For the germline transmission experiments, it is not clearly mentioned how many founder fish was tested and how the mosaicism of (germline of) EACH founder fish was. This is important.

(2) Explain in the legends that what the numbers on the top of the gels mean. For instance, the gel images in Supp Fig. 5, 9. 19. 21.

(3) In Discussion,

However, current KI methods have limitations restricting their applications in some studies

The original MMEJ-mediated KI strategy is an excellent method in principle but showed variations in KI efficiencies that cannot be ignored in real use

I admire the authors' comprehensive analyses, but these are too much disgrading of other works. Other systems also DID work and the authors' system will not be the only one to be carried out for KI since the purpose of KI will differ in each experiment.

Reviewer #2 (Remarks to the Author):

In this paper, the authors examined (1) knock-in with MMEJ using a single sgRNA, (2) a fluorescence reporting method for low-expression genes by applying Gal4-UAS, and (3) knock-in with chemically modified linearized donor DNA. The knock-in efficiency has been evaluated mainly on the basis of fluorescence observation and junction PCR, and certain results have been obtained. In particular, the fact that the knock-in efficiency was evaluated at 31 loci is worthy of evaluation. However, I do not recommend that this paper be published in Communications Biology due to the following several critical issues.

Major Concerns.

1. The importance of fluorescent reporting of gene expression is clearly declining as single cell RNA-seq is becoming mainstream. On the other hand, Protein-tagging, as represented by Bio-ID, is gaining importance. Unfortunately, the authors' direction focuses only on gene-tagging, and I conclude that this paper does not have much importance.

2.

Since the S-NGG-25 method of the authors even knocks in the backbone of the donor (Fig. 1 A1), no targeted endogenous gene is expressed from the knock-in allele. It is also fully expected that the indel mutation will be introduced into the opposite allele. This means that the S-NGG-25 method cannot be used for the more important genes that are essential for embryonic development. In addition, what

confused me was the authors' RT-qPCR data. The information on gene structure and the site where Primer anneals is also unknown, and it was shown that the expression of endogenous genes is elevated in KI over WT, like cx55.5, which is completely contradictory to the authors' knock-in strategy.

3.

It had not been examined whether the expression pattern of the fluorescent reporter gene mimicked that of the target endogenous gene.

4.

Survival and developmental rates of embryos in which genome editing factors were introduced are not described, so the practical usefulness of this method is unclear.

5.

Despite the possibility that fluorescent reporter genes may not be expressed from random integration alleles, no data are presented confirming the random integration of chemically modified linearized donor DNA by genomic PCR.

Reviewer #3 (Remarks to the Author):

This manuscript contains a very interesting study describing clever knock-in methods with some novel aspects. The most useful and novel part is the amplification strategy (especially the VH-mediated IFL strategy) making it possible to target lowly expressed genes. I have a few points to address before it may be published:

1. It is important the authors describe the advantages of this method more clearly (especially over the GeneWeld method that is currently most commonly used).

For the in frame KI the advantages include:

A. correct integration can be inferred by that the expression pattern of the FP is mimicking the endogenous gene's expression pattern (but not with other methods relying on eye or heart markers).

For the 5' modified dsDNA the advantages include:

A. no need to clone in vivo linearization sites (not just "to avoid inserting unwanted plasmid fragments into the genome").

B. increased efficiency of germline transmission and accuracy.

For the single linearization antisense site this might include other things...

2. A preceding similar 3' knock-in method with 5' modified donors has recently been published by Mi et al. Life Science Alliance, and should be duly cited and discussed.

3. I don't understand what the "Circular or linearized CMV-eGFP donors" are (Fig 4D), and why the circular dsDNA seems most efficient (even more than the 5' modified dsDNA donors). Please, explain.

4. Define "IamGolden" and consider removing the term from the abstract as it is not that commonly known.

5. Did the authors use 5'-chemically modified linear donors to generate lines with the amplified expression? If so, please highlight the advantage of the combination of their approaches. In my

opinion, the combination of the VH-mediated IFL strategy with PCR amplification using primers with 5' modifications and harbouring short homology arms, would be the most useful contribution of this paper. The VH-vector also carrying a 5×nrUAS-E1b-eGFP displayed in Fig 3 could in this way serve as a template for any gene. Therefore, it would be useful for the zebrafish community if the authors deposited this vector (and other useful ones) to Addgene.

6. In the schematic for VH-mediated IFL strategy the colour of the HSF1 is yellow in the box but brown in the protein. Please harmonize (and double-check colour-codes throughout).

Point-to-point response to reviewers' comments:

Reviewer #1 (Remarks to the Author):

In the manuscript, the authors performed experiments to knock-in donor DNAs to the zebrafish genome via the CRISPR/Cas9 method. They tested many conditions to optimize the frequency. Namely, the number of cutting sites, the length of the homology arm, signal amplification via Gal4-UAS system, modification of the linearized donor, etc. Also, the authors applied this system to create KI in authors Cx genes and demonstrated this approach is feasible. Although the work itself is not perfectly innovative, the authors conducted comprehensive analyses and the information provided is useful to all researchers who are planning to do similar KI experiments.

Response: We thank the reviewer very much for positively commenting on our work.

I have the following comments:

(1) For the germline transmission experiments, it is not clearly mentioned how many founder fish was tested and how the mosaicism of (germline of) EACH founder fish was. This is important.

Response: We thank the Reviewer for the advice. The number of tested founder fish in Fig. 1e, 2d, and 5d of the revised manuscript (Fig. 1e, Supplementary Fig. 12, and Fig. 4f in the original manuscript) and the mosaicism of each founder fish were summarized in Supplementary Data 5 in the revised manuscript. Figures showing the mosaicism of each founder fish were added in Fig. 1g, 2e, and Supplementary Fig. 25c in the revised manuscript (Line 118-120, Line 147-149, and Line 263-264) and were shown below.

Fig. 1g in the revised manuscript

Fig. 2e in the revised manuscript

Supplementary Fig. 25c in the revised manuscript

(2) Explain in the legends that what the numbers on the top of the gels mean. For instance, the gel images in Supp Fig. 5, 9, 19, 21.

Response: We thank the Reviewer for pointing out the missing information. We apologize for this carelessness and have added the missing information in the corresponding places in the legends. For example, in Supplementary Fig. 9, 12, 20, 21, and 25 in the revised manuscript (Supplementary Fig. 9, 5, 19, 21, and 23 in the original manuscript), the gel images are genotyping results for F₁ zebrafish and the numbers on the top are the numbers of the F₀ fish from which the F₁ were generated. “+” means GFP-positive and “-” means GFP-negative.

(3) In Discussion,

However, current KI methods have limitations restricting their applications in some studies.

The original MMEJ-mediated KI strategy is an excellent method in principle but showed variations in KI efficiencies that cannot be ignored in real use.

I admire the authors' comprehensive analyses, but these are too much disgrading of other works. Other systems also DID work and the authors' system will not be the only one to be carried out for KI since the purpose of KI will differ in each experiment.

Response: The Reviewer is right that our comment is inappropriate. We re-wrote this part as “To supplement the current gene-editing toolbox, here through several optimizations we reported an improved MMEJ-based KI strategy for efficient and accurate fluorescence-

labeling of almost every gene of interest in zebrafish, whether it is endogenously highly expressed or not.” (Line 278-282) and “In comparison with current MMEJ-mediated KI strategies, including the GeneWeld, the method reported here has several improvements.” (Line 283-284).

Reviewer #2 (Remarks to the Author):

In this paper, the authors examined (1) knock-in with MMEJ using a single sgRNA, (2) a fluorescence reporting method for low-expression genes by applying Gal4-UAS, and (3) knock-in with chemically modified linearized donor DNA. The knock-in efficiency has been evaluated mainly on the basis of fluorescence observation and junction PCR, and certain results have been obtained. In particular, the fact that the knock-in efficiency was evaluated at 31 loci is worthy of evaluation. However, I do not recommend that this paper be published in Communications Biology due to the following several critical issues.

Major Concerns.

1. The importance of fluorescent reporting of gene expression is clearly declining as single cell RNA-seq is becoming mainstream. On the other hand, Protein-tagging, as represented by Bio-ID, is gaining importance. Unfortunately, the authors' direction focuses only on gene-tagging, and I conclude that this paper does not have much importance.

Response: We thank the Reviewer for the comment. We agree that single cell RNA-seq is a very powerful tool. But gene-tagging, scRNA-seq, and Bio-ID are different techniques for different purposes. These techniques complement each other in addressing biological questions. For example, scRNA-seq reveals gene expressions at the RNA level, but the RNA level is not equal to the protein level and scRNA-seq cannot address issues related to real-time *in-vivo* imaging and tracking. Similarly, Bio-ID (proximity-dependent biotin identification) is a good tool for studying protein-protein interactions, but it is not suitable for real-time detection and localization of proteins of interest *in vivo*. A wide application of gene-tagging techniques is unavoidable. One can achieve multiple purposes using the gene-tagging system, including *in vivo* analyzing, imaging, and tracking real-time expression of genes of interest and identifying interactomes of the tagged gene/protein in pulldown assays.

2. (The first half of this comment) Since the S-NGG-25 method of the authors even knocks in the backbone of the donor (Fig. 1 A1), no targeted endogenous gene is expressed from the knock-in allele. It is also fully expected that the indel mutation will be introduced into the opposite allele. This means that the S-NGG-25 method cannot be used for the more important genes that are essential for embryonic development.

Response: We thank the Reviewer for the comment. We are sorry that we do not know whether we fully understand these criticisms, but we tried to respond to these questions in the way we understand them.

(1) The reviewer is right that part of the backbone of the donor plasmid downstream of the polyA signals was knocked in at the 3'-junction by the S-NGG-25 method, but we do not understand how the reviewer concluded that "no targeted endogenous gene is expressed from the knock-in allele". As shown by data in Supplementary Fig. 9, 12, and 25b in the revised manuscript (Supplementary Fig. 9, 5, and 23b in the original manuscript), seamless knock-in was achieved and the CDS provided by the donor plasmid ensured that endogenous expression of the target gene was not interrupted.

(2) The reviewer raised an essential question that "the indel mutation will be introduced into the opposite allele". The reviewer is right that indel mutation(s) would occur with a high

probability positively correlated with the high efficiency of the KI. This issue applies to all CRISPR/Cas9-mediated recombination methods. We all know that backcross to the wild-type background is commonly used to remove the indel mutations on the other allele. Since the fish with lethal indel mutation(s) should be selected out when live F₀ fish were collected, we do not think the occurrence of these mutations in some fish would limit the application of the S-NGG-25 method.

To address the comment that due to the indel mutation(s), “the S-NGG-25 method cannot be used for the more important genes that are essential for embryonic development”, we generated zebrafish with GFP-tagged *tbx5a* and *tnni1b*, respectively. Zebrafish *tbx5a* is closely associated with both the pectoral fin development and heart regeneration and the loss of one allele results in the absence of one pectoral fin (PMID: 12066188, 29933372, 29382818). Deficiency of *tnni1b* results in severe pericardial edema, malformation of the heart tube, and about 88% of lethality at 7 dpf (PMID: 30044923). After microinjection of Cas9 mRNA, *tbx5a* sgRNA or *tnni1b* sgRNA, *lamGolden* sgRNA, and S-NGG-25 based KI donor into one-cell-stage embryos, low survival rates of F₀ zebrafish (about 30% for *tbx5a* and 15% for *tnni1b*) were observed. F₀ zebrafish with pectoral fin deficiency for *tbx5a* or heart malformation for *tnni1b* (as shown below and in Fig. 3d in the revised manuscript) were found, supporting that these genes are crucial for zebrafish embryonic development. Nevertheless, GFP-chimeric F₀ zebrafish were obtained and healthy *tbx5a*^{+/+eGFP} and *tnni1b*^{+/+eGFP} F₁ zebrafish were further identified. In consistence with the literature (PMID: 31663848, 30044923), *tbx5a* was expressed in the pectoral fin and heart while *tnni1b* expression was only detectable in the heart (as shown below and in Fig. 3e and f in the revised manuscript). These results indicated that the S-NGG-25 KI strategy can be used for genes critical for embryonic development. The reason for this is not that lethal indel mutations cannot be generated but that our KI procedures selected those live F₀ fish with no lethal mutations. We provided this information in the revised manuscript (Line 171-178).

Fig. 3d in the revised manuscript

Fig. 3e and 3f in the revised manuscript

2. (The second half of this comment) In addition, what confused me was the authors' RT-qPCR data. The information on gene structure and the site where Primer anneals is also unknown, and it was shown that the expression of endogenous genes is elevated in KI over WT, like *cx55.5*, which is completely contradictory to the authors' knock-in strategy.

Response: We apologize for the missing information. Gene structure and qPCR primer binding sites are provided in the figure below. Detailed methods were described in the Materials & Methods section (including the step of adding genomic DNA wiper), and qPCR primer sequences were summarized in Supplementary Data 3. Regarding the expression levels of *connexin* genes in WT and KI zebrafish, our data showed that KI had no influence on most of them. The biggest difference (about 2-fold) was observed between WT and the *cx55.5* KI zebrafish. Such influence is most likely case-dependent and we do not think it is “completely contradictory to the authors' knock-in strategy”.

*Gene structure and qPCR primer binding sites for Supplementary Fig. 15c in the revised manuscript (Supplementary Fig. 14c in the original manuscript)

3. It had not been examined whether the expression pattern of the fluorescent reporter gene mimicked that of the target endogenous gene.

Response: The reviewer is right that it would be necessary to confirm “whether the expression pattern of the fluorescent reporter gene mimicked that of the target endogenous gene”. We compared the GFP expression patterns of our KI zebrafish with the published *in-situ* hybridization results (data sources were included in the figures below) for some *connexins*, and found that the patterns are the same (Figures shown below). For example, *cx23*, *cx44.1*, and *cx48.5* are only expressed in the lens. *cx35* is mainly expressed in the spinal cord. *cx39.9* is expressed in the muscle. *cx43* is expressed in the cerebral cortex,

notochord, spinal cord, lens capsule, and ultimobranchial body, and *cx52.6* is expressed in the brain.

*Images showing GFP expression patterns of the KI zebrafish generated by this work (left panel) and the corresponding published *in situ* hybridization results (right panel).

4. Survival and developmental rates of embryos in which genome editing factors were introduced are not described, so the practical usefulness of this method is unclear.

Response: We took *cx43* and *cx43.4* as examples. Wild-type embryos were injected with NFW (nuclease-free water), KO (Cas9 mRNA and sgRNA), or KI (Cas9 mRNA, two sgRNAs, and donor) cocktails and the survival rates were recorded. As shown below and in Supplementary Fig. 5 in the revised manuscript, survival rates decreased with the increase of the amounts/complexity of the injection components, suggesting that the physical injury caused by microinjection and the toxicity caused by gene-editing materials did exist. However, the survival rates of the *cx43* KI and *cx43.4* KI groups were both above 50%, therefore, cytotoxicity of the KI procedure will not affect the establishment of the KI germline.

As for the 35 KI fish lines we generated, the survival and developmental rates of their embryos are comparable with those of the wild-type fish. We provided this information in the revised manuscript (Line 122-124).

Supplementary Fig. 5 in the revised manuscript

5. Despite the possibility that fluorescent reporter genes may not be expressed from random integration alleles, no data are presented confirming the random integration of chemically modified linearized donor DNA by genomic PCR.

Response: We thank the Reviewer and performed experiments to check the possibilities of random integration as suggested. We used the published Cyclic Digestion and Ligation-Mediated PCR (CDL-PCR) method (PMID: 32103092) for the *cx43*^{+linear-eGFP/+linear-eGFP} and *cx43.4*^{+linear-eGFP/+linear-eGFP} KI zebrafish (as shown below and in Supplementary Fig. 26 in the revised manuscript). BclI- or BglII-digested genomic DNA was ligated with Sau3AI-digested adaptor by T4 DNA ligase. NcoI-, BspHI-, or PciI-digested genomic DNA was ligated with FatI-digested adaptor by T4 DNA ligase. Then nested PCR was performed to detect random integration. Sequencing analysis of the PCR products showed that the modified linearized dsDNA donors were integrated into the desired target sites for both genes, and no random integration was detected. We provided this information in the revised manuscript (Line 267-273).

Supplementary Fig. 26 in the revised manuscript

e Nested PCR for cx43

Sequence blast of BspHI-F3

Danio rerio connexin 43, mRNA (cDNA clone MGC:55225 IMAGE:5915503), complete cds
 Sequence ID: BC049297.1 Length: 2821 Number of Matches: 1

Range 1: 1395 to 1675 GenBank Graphics Next Match Previous Match

Score	Expect	Identities	Gaps	Strand
520 bits(281)	1e-142	281/281(100%)	0/281(0%)	Plus/Plus
Query 1	GCAGAGTACATGCCCTAACTACCGAACAGTCAAGGGGGCTCGTCCGGGACGAACGGCA	60		
Sbjct 1395	GCAGAGTACATGCCCTAACTACCGAACAGTCAAGGGGGCTCGTCCGGGACGAACGGCA	1454		
Query 61	CTGAACCTGCCAGTCTCTCAACTCAGCCACCAGAACAGACTTGGCGATGTGATGAT	120		
Sbjct 1455	CTGAACCTGCCAGTCTCTCAACTCAGCCACCAGAACAGACTTGGCGATGTGATGAT	1514		
Query 121	TTTTGTGTGCTTGTGAAATGCCACAAATGATCCACTTTAACACTTGGACTCTACT	180		
Sbjct 1515	TTTTGTGTGCTTGTGAAATGCCACAAATGATCCACTTTAACACTTGGACTCTACT	1574		
Query 181	AGTTTGTGTAGATTGTGTCTAAccccccTACAGTCGATCCCGGTATAAGTCAATC	240		
Sbjct 1575	AGTTTGTGTAGATTGTGTCTAAccccccTACAGTCGATCCCGGTATAAGTCAATC	1634		
Query 241	GTTCCATGATCTAAATCCACAGGGGTTCACAGTCATG	281		
Sbjct 1635	GTTCCATGATCTAAATCCACAGGGGTTCACAGTCATG	1675		

f Nested PCR for *cx43.4*

Sequence blast of BclI-R3

Danio rerio connexin 43.4, mRNA (cDNA clone MGC:191652 IMAGE:100059961), complete cds
 Sequence ID: BC164477.1 Length: 1210 Number of Matches: 1

Range 1: 385 to 684 GenBank Graphics ▼ Next Match ▲ Previous Match

Score	Expect	Identities	Gaps	Strand
555 bits(300)	3e-153	300/300(100%)	0/300(0%)	Plus/Plus
Query 1	GATCAACCGCGGACCAACCGGGATTATGAGGAGCGGAGACAAACGGTAGGAGATCC	60		
Sbjct 385	GATCAACCGCGGACCAACCGGGATTATGAGGAGCGGAGACAAACGGTAGGAGATCC	444		
Query 61	TATGATTATGGAGAGATCGTCCGAGAAAGAAAGGCTCCAGAGAGCTCGCTGTAA	120		
Sbjct 445	TATGATTATGGAGAGATCGTCCGAGAAAGAAAGGCTCCAGAGAGCTCGCTGTAA	504		
Query 121	ACATGACCGCGCGGAGAAATAAAGCGAGATGGGCTCATGAGGTGTACATCTCGAGCT	180		
Sbjct 505	ACATGACCGCGCGGAGAAATAAAGCGAGATGGGCTCATGAGGTGTACATCTCGAGCT	564		
Query 181	TCTGTCGAGGATATTTTCGAGGTGGGTTTCCTCTTGGCCAGTATATCTGTATGGTTT	240		
Sbjct 565	TCTGTCGAGGATATTTTCGAGGTGGGTTTCCTCTTGGCCAGTATATCTGTATGGTTT	624		
Query 241	CGAGGTCCCGCGTACATCGTGTGACTCGAGTCCCTGCCCGCACCGTAGACTGCTT	300		
Sbjct 625	CGAGGTCCCGCGTACATCGTGTGACTCGAGTCCCTGCCCGCACCGTAGACTGCTT	684		

Sequence blast of BspHI-R3

Danio rerio connexin 43.4, mRNA (cDNA clone MGC:191652 IMAGE:100059961), complete cds
 Sequence ID: BC164477.1 Length: 1210 Number of Matches: 1

Range 1: 541 to 840 GenBank Graphics ▼ Next Match ▲ Previous Match

Score	Expect	Identities	Gaps	Strand
555 bits(300)	3e-153	300/300(100%)	0/300(0%)	Plus/Plus
Query 1	CATGAAGGTGTACATCTCGAGCTTCTGTCGAGGATATTTTCGAGGTGGGTTTCCTCT	60		
Sbjct 541	CATGAAGGTGTACATCTCGAGCTTCTGTCGAGGATATTTTCGAGGTGGGTTTCCTCT	600		
Query 61	TGGCCAGTATATCTGTATGGTTTCGAGGTGGGCGGTCATACGTGTGACTCGAGTCC	120		
Sbjct 601	TGGCCAGTATATCTGTATGGTTTCGAGGTGGGCGGTCATACGTGTGACTCGAGTCC	660		
Query 121	CTGCCCCACACCGTAGACTGCTTGTGTCACTCCGACAGAGAAACCACTTCTCTGCT	180		
Sbjct 661	CTGCCCCACACCGTAGACTGCTTGTGTCACTCCGACAGAGAAACCACTTCTCTGCT	720		
Query 181	GATTATGATGCCGTGAGCTGTCTGCTGTCTTACGGTGTGGAGATCTTCATT	240		
Sbjct 721	GATTATGATGCCGTGAGCTGTCTGCTGTCTTACGGTGTGGAGATCTTCATT	780		
Query 241	GGGCCCTCAGCGGATTCGTGATGCTTTTCGACGACGTCACGCCATCAAGGTTCAGCG	300		
Sbjct 781	GGGCCCTCAGCGGATTCGTGATGCTTTTCGACGACGTCACGCCATCAAGGTTCAGCG	840		

Reviewer #3 (Remarks to the Author):

This manuscript contains a very interesting study describing clever knock-in methods with some novel aspects. The most useful and novel part is the amplification strategy (especially the VH-mediated IFL strategy) making it possible to target lowly expressed genes. I have a few points to address before it may be published:

1. It is important the authors describe the advantages of this method more clearly (especially over the GeneWeld method that is currently most commonly used).

For the in frame KI the advantages include:

A. correct integration can be inferred by that the expression pattern of the FP is mimicking the endogenous gene's expression pattern (but not with other methods relying on eye or heart markers).

For the 5' modified dsDNA the advantages include:

A. no need to clone in vivo linearization sites (not just "to avoid inserting unwanted plasmid fragments into the genome").

B. increased efficiency of germline transmission and accuracy.

For the single linearization antisense site this might include other things...

Response: We thank the Reviewer very much for the advice and positive comments. We have rewritten the Discussion of our manuscript to describe the advantages of our method more clearly (Line 283-314).

2. A preceding similar 3' knock-in method with 5' modified donors has recently been published by Mi et al. Life Science Alliance, and should be duly cited and discussed.

Response: Thanks for providing us with this information. We cited and discussed this work in the revised manuscript (Line 315-318).

3. I don't understand what the "Circular or linearized CMV-eGFP donors" are (Fig 4D), and why the circular dsDNA seems most efficient (even more than the "Circular or linearized CMV-eGFP donors" donors). Please, explain.

Response: We apologize for the confusion. "Circular or linearized CMV-eGFP donors" are different forms of dsDNAs containing *CMV-eGFP* (Supplementary Fig. 23b in the revised manuscript (Supplementary Fig. 22a in the original manuscript)), which were used as reporters to examine the protective functions of chemical modifications on dsDNA from degradation. To make it clearer, we revised the description as "Consistently, using a *CMV-eGFP* mini reporter, we proved that 5'-four phosphorothioate (5'-4PS)-modified⁵⁶ dsDNA gave significantly higher GFP expression than the unmodified linearized dsDNA, while the circular dsDNA provided the highest GFP expression" (Line 245-248 in the revised manuscript).

In Fig. 5b and c of the revised manuscript (Fig. 4d and e in the original manuscript), the circular dsDNA donor and the linearized dsDNA donors were all constructed based on the S-NGG-25 method (Fig. 5a in the revised manuscript). The major difference between

them was their existing forms, linearized or circular, modified or unmodified. One possible explanation for the phenomenon that the circular dsDNA was more efficient than the linearized dsDNAs is that the circular dsDNA was more stable in cells than the linearized dsDNAs, which was supported by the results in Supplementary Fig. 23b and c in the revised manuscript (Supplementary Fig. 22a and b in the original manuscript). To further test this possibility, we injected equal amounts (0.9×10^{-17} mol per embryo) of unmodified dsDNA donor, 12PS-modified dsDNA donor, and circular plasmid donor into one-cell-stage embryos, respectively. Five embryos were randomly selected and mixed for each group to extract total DNA at different time points. PCR analysis showed that the unmodified linearized dsDNA was less stable than the 12PS-modified dsDNA and was not detectable at 48 hpf. In contrast to the time-dependent gradual degradation of both of the linearized dsDNAs, circular plasmid hardly showed any sign of degradation within 24 hpf and was still at a high abundance at 48 hpf (data shown below and in Supplementary Fig. 24 in the revised manuscript). We provided this information in the revised manuscript (Line 255-257).

Supplementary Fig. 24 in the revised manuscript

4. Define "lamGolden" and consider removing the term from the abstract as it is not that commonly known.

Response: Thanks and we rewrote related sentences as suggested in the abstract (Line 23-24, Line 83).

5. Did the authors use 5'-chemically modified linear donors to generate lines with the amplified expression? If so, please highlight the advantage of the combination of their approaches. In my opinion, the combination of the VH-mediated IFL strategy with PCR amplification using primers with 5' modifications and harbouring short homology arms, would be the most useful contribution of this paper. The VH-vector also carrying a 5xnrUAS-E1b-eGFP displayed in Fig 3 could in this way serve as a template for any gene.

Therefore, it would be useful for the zebrafish community if the authors deposited this vector (and other useful ones) to Addgene.

Response: As suggested, we will make our plasmids available upon request and deposit them in Addgene.

We agree with the reviewer that a combined strategy could be more powerful and we tried to combine the VH strategy and the 5'-chemically-modified linear donor-mediated KI strategy to generate KI zebrafish for *cx34.4*, *cx43.4*, and *cx47.1* (Data below and in Supplementary Fig. 27 in the revised manuscript). However, we found that the ratios of PCR-positive F_0 were very low, even though several PCR-positive F_0 embryos were obtained for each gene. For *cx34.4* and *cx43.4*, junction PCR and GFP expression patterns confirmed that the 12PS-modified linearized dsDNA was precisely integrated into the target site and the *VH-eGFP*-tagged F_1 were generated. But for *cx47.1*, we didn't obtain *VH-eGFP*-tagged F_1 zebrafish from 10 PCR-positive F_0 . These results suggested that the 5'-chemically modified linearized VH-adapted donor-mediated KI strategy was doable/feasible but the KI efficiency was low and further optimizations were needed if it was to be applied to all genes. We provided this information in the Discussion of the revised manuscript (Line 326-337).

Supplementary Fig. 27 in the revised manuscript

6. *In the schematic for VH-mediated IFL strategy the colour of the HSF1 is yellow in the box but brown in the protein. Please harmonize (and double-check colour-codes throughout).*

Response: We are sorry and have made the color codes consistent throughout our manuscript in the revision (for example, Fig. 4b in the revised manuscript (Fig. 3a in the original manuscript)).

Reviewers' comments:

Reviewer #1 (Remarks to the Author):

I think the authors responded to my comments promptly.

Reviewer #2 (Remarks to the Author):

Kindly confirm the attached file.

Reviewer #3 (Remarks to the Author):

The authors have satisfactorily answered to my questions, and I suggest it now should be published